# Factors Affecting the Sustainable Development of HRS in Transforming Economies: A fsQCA Approach

**Wen Chen** [1] , **Xiao-Jiao Song** [2] **and Yanping Li** [1,*]

1 School of Business and Management, Wuhan University, Wuhan 430072, China; 2018101050089@whu.edu.cn
2 School of Criminal Science and Technology, Nanjing Forest Police College, Nanjing 210023, China; songxiao607@126.com
* Correspondence: ypli@whu.edu.cn; Tel.: +86-027-6875-3084

**Abstract:** Human resources services (HRS) are kinds of services concerning human resources acquisition, development, and allocation provided to employers and workers. The services promote the efficiency and effectiveness of the human resources market. Recently, the services have been regarded as crucial for the sustainable development of the national economy, attracting policymakers in transforming economies to promote the expansions of the services. This paper presents a systematic study of factors that affect the development of the services. In this text, the fuzzy-set qualitative comparative analysis (fsQCA) method was used to explore the configurations of the factors that drive the growth of the services. The data was from province governments' statistics and influential research reports in China, and each province observation was considered as a case (in fsQCA terms). In this article, the marketization of human resources (MOHR) referring to allocating human resources through the labor market rather than government bureaucracy, is a possible necessary condition for the boom of the HRS. Moreover, we identified seven paths to explain both the high and low development level of HRS. Among the paths, the configuration of general development of regional economy, social legitimacy of the services, and marketization of human resources are the basic conditions that lead to a high growth rate of the HRS. These research findings enriched our understanding of the valid strategies for cultivating the services industry in China and other transforming countries. To cultivate and upgrade the services, we suggested that transforming economies should pay more attention to economic internationalization and speed up the marketization of human resources.

**Keywords:** human resources services; transforming economy; fsQCA approach; marketization of human resources; sustainability

## 1. Introduction

Human resources services (HRS) are kinds of services concerning human resources acquisition, development, and allocation provided to employers and workers. The services promote the efficiency and the effectiveness of the human resources market. According to the economic reports of the World Employment Confederation in 2019, there are many kinds of the services, such as private employment service, agency work, management service providers, direct recruitment, recruitment process outsourcing, and career management. Researchers have found that the services began in the Midwestern United States in the 1920s, and have expanded to Western Europe, Australia, East Asia, Eastern Europe and other areas [1–3]. Recently, the services became more important for the human resources market, with the advent of the Fourth Industrial Revolution and fiercer international competition. To achieve a high-quality (with regards to sustainability, innovation-driven, and services dominance) economic development and full employment, China and other transforming economies are promoting the development of the HRS.

However, little research has explored how the configurations of factors, such as institutional environment [4], economic internationalization [5], and the employable population

affect the development of the services systematically. The current research showed that the services not only promote regional employment by reducing labor market transaction costs [6–8], but also create numerous jobs on the services [9]. The services have evolved into an industrial sector. In the last few years, the human resources services industry (HRSI) has developed rapidly even after the financial crisis of 2008. In the post-COVID-19 era, the services play vital roles in promoting enterprises to resume work and accelerating the economic recovery of China. The services have attracted policymakers to promote the sustainable development of such professional service [10,11]. There are lots of public policies driving the expansion of the services. Unfortunately, the dim causality between the factors and the development of the services has led to half the result with twice the effort for the cultivation of the services in transforming economies. The fuzzy-set qualitative comparative analysis (fsQCA) [12] was used to explore the causal logic.

The work of this paper revealed the importance of the marketization of human resources for the HRS and enriched our understanding of the valid strategies of regional governments to cultivate the industry in China and other transforming economies.

The structure of this paper is as follows. Section 2 explains the theoretical hypotheses. Section 3 presents a method, data, and initial calibration. Section 4 provides the results and fsQCA analysis. Section 5 contains the contributions and implications of the findings.

## 2. Theoretical Framework and Hypotheses

HRS agencies are active in the HRS ecology [13–17]. Among the ecology, there are some major factors that affect the sustainable development of HRS. Based on a literature review on HRS and combining the growth of the services in China, we found the factors and formed five theoretical assumptions as follows.

The first is that the regional economic boom creates favorable conditions for the development of the HRSI [9,18]. On one hand, regional flourishing economies generally create more jobs, which attract job hunters and employers to decrease transaction costs in the human resources market by using the HRS [8,19]. On the other hand, regional development potentially provides employers with more opportunities to realize scale economy and the specialization of production, which should contribute to generating more demand for professional HRS. As a result, the regional governments could promote the development of HRSI from the demand side [20,21]. It follows that there should be positive relations between the regional economy and the sustainable development of HRSI.

**Hypothesis 1 (H1).** *The regional economic volume favors the development of HRSI in the same region.*

The second is that the quantity and quality of human resources in a region affect the development of HRSI. It is generally believed that workers tend to get employed through the human resources market because of the capitalist spirit of market competition [22]. Additionally, in China and other Asian regions, where Confucianism is popular, leaving their hometowns not only means the economic cost of travel, but also refers to separation from family, which tends to result in social anxiety [23]. Usually, laborers have little intention to relocate [24]. Furthermore, the free flow of the employable across regions is restricted from unequal opportunities to better education, health services, and other public services related to household registration system in China [24]. The employable, especially laborers with little income, tend to flow from less developed regions to the cities of developed provinces for employment. To maintain a work–family balance, they commute between their working place and household registration location regularly. When there are efficient jobs in their hometown, they tend to return and use the HRS to hunt for jobs. Therefore, we get a proposition as follows.

**Hypothesis 2 (H2).** *The size of regional the employable population contributes to the development of HRSI.*

The third is that the internationalization of economies accelerates the development of HRSI. It is well known that HRS has developed for decades in most developed countries, such as the U.S., the U.K., and Canada. Due to a high degree of maturity in the services market, HRS agencies in these countries are facing intense competition and falling marginal revenue of the services. To disperse business risks and obtain more profits, a few HRS enterprises follow the internationalization of their customers and expand their business in host countries [11,25]. These enterprises have become a significant part of internationlization [26]. Additionally, the regional economic and political development, such as the enlargement of the European Union, has provided new opportunities for the enterprises [11,27,28]. Also, some HRS agencies have promoted the flow of the employable population from the underdeveloped to developed areas [29,30]. Moreover, the expansion of multinational HRS enterprises promotes the development of HRSI in host countries. Unsurprisingly, the expansion of multinational HRS enterprises such as Delco, Randstad, and Manpower introduced new business models, and cultivated professional teams of HRSI for China [31]. Therefore, we presume the importance of foreign investment from developed countries for the growth of the HRSI in host areas.

**Hypothesis 3 (H3)**. *The foreign direct investment (FDI) from developed centuries promotes the sustainable development of HRSI in transforming economies greatly.*

The fourth hypothesis is that the marketization of human resources is of great importance for the realization of the value of the employable. The human resources market affects the effective allocation of human resources as mentioned above. Meanwhile, the forming of the market is different between developed areas and transforming economies. As everyone knows, markets are evolving with the development of capitalism in the U.S., the U.K., and other free capitalist countries naturally [32,33]. However, the cultivation of the market in China and other transforming economies is attributed to market-oriented employment system reform [31]. In the planned economy system, laborers participated in the job according to the arrangement of the government employment department. Recently, along with the transforming of the employment system, both workers and employers participate in the labor market gradually and allocate human resources by the market rather than the government. The development of a unified human resources market is still a work in progress in China and other emerging economies. The marketization index of human resources refers to the progress of the relationship between government and the services market, the development of non-state HRS providers, and the construction of legal system environment for HRS, etc. There are various levels of the marketization index of human resources in regions due to different progress of market-oriented reform and opening in China. Consequently, we assume another proposition as follows.

**Hypothesis 4 (H4)**. *The marketization of human resources (MOHR) affects the sustainable development level of HRSI positively.*

Finally, the core aim of the reform in the human resources market in China is to provide allocation efficiency and flexibility for the market. Generally speaking, the HRS is a double-edged sword, which increases the employment opportunities for job hunters and possibly deteriorates the employment security of workers at the same time. People have different attitudes towards HRSI due to different employment culture and traditions [34,35]. For example, in the U.S. and U.K., laborers naturally accept the marketization and flexibility of human resources. While, in German and East European countries, it is difficult to get the flexibility of human resources because of powerful negotiation from labor unions [13,16,35]. The Hartz reform in German finally deregulated the HRSI with great difficulty [36]. Suppose that a society is a community, the mainstream of public opinion represents the social legitimacy for a particular thing or phenomenon. China's government serves all sectors of society. Public policies could better represent the attitudes of the community towards a particular issue in China. The higher the social legitimacy of a particular services, the

smaller the social resistance to the expansion of the services. To obtain legitimacy, the HRS agents try to show their roles in promoting the flexibility of human resources, and exercise self-discipline to avoid the infringement on the workers during the reform [14,37]. Moreover, the development of HRSI is also significantly affected by the legal construction of China. Therefore, we grant a hypothesis as follows.

**Hypothesis 5 (H5)**. *The social legitimacy (SLEG) of HRSI positively influences the sustainable development of HRSI.*

In this paper, we examine the five conditions mentioned above working in a combined way to explain the development of regional HRSI in transforming economies. As noted above, we have proposed theoretical assumptions about individual conditions affecting the development of HRSI. However, it is important to notice that the theoretical assumptions about individual conditions could be neither confirmed nor disconfirmed in fsQCA, since the influence of individual conditions on the outcome of interests relies on their context. Furthermore, the main function of theoretical assumptions in fsQCA is to help us identify the necessary and sufficient conditions for a specific outcome. Also, the necessity and sufficiency based on the deterministic logic is the key causality examined in QCA, while the correlation relationship based on the probabilistic logic is not of interest in this study.

## 3. Method, Data, and Initial Calibration

This section presents basic information about the method used in this context, the data considered, the preliminary calibration of conditions, and the outcome (required for use with fsQCA).

### 3.1. Method

This research used the Qualitative Comparative Analysis (QCA) method to perform the analysis. Because there is a good quality of data, we can show the fine-grained differences among cases. There are three types of QCA: crispQCA, multivalueQCA, and fsQCA [12]. As a technique that goes beyond qualitative and quantitative research methods, fsQCA is increasing popular in business and social sciences research [38]. Compared to the other two kinds of QCA, fsQCA allows continuous variables in the models. It follows an inductive approach [39], in this case, finding configurational relationships between the conditions (many factors) and outcome (the development level of HRSI). After deciding to apply fsQCA, we firstly calibrated the data, and then conducted a necessity analysis. Consequently, we formulated truth tables. Then, we analyzed the necessary and sufficient conditions. Finally, we interpreted the solution formula. In this article, the R4-0-3 QCA package was used to conduct the analysis and to identify paths that led to the outcome.

### 3.2. Data

To achieve the goal of this research, we followed the requirements of the fsQCA method. Firstly, we considered the limitation of samples, regarding the largest similarity of sample attributes and the largest difference in sample results [12]. Then, we took into account the availability of data. Finally, we selected China as the source of cases. China has the largest transforming economy and second-largest economy in the world. The central government has been promoting the development and upgrading of the HRSI especially after the 18th Congress of the Communist Party of China. The provinces of China obey the unified planning of the central government. Meanwhile, the development levels of HRSI are different between the provinces, which created convenient conditions for our research. Next, we report the source of the data used in the research.

1.  **GDP.** There are a lot of ways to measure the volume of the regional economy, while the most popular indicators used are the gross national product (GNP) and the gross domestic product (GDP). Considering China's opening to the world and there is a

greater degree of internationalization, we believe that the GDP fits our research better than GNP. As a result, we chose GDP to survey the volume.

2. **PLR**. There was no accurate data of the employable population of each province until now. Furthermore, neither the existing registered population nor employment quantity can reflect this indicator. Regarding the characteristics of China's population mobility and employment habits, the population of long-term residents (PLR) should briefly characterize the scale of the employable in a region. Therefore, we used the value of long-term residents as the indicator alternatively.

3. **FDI**. In the past decades, China has further expanded its opening, striving to build a community with a shared future for mankind. Chinese people have benefited from the reforms. Recently, the reforms have entered deep water zone. All the provinces attach greater importance to the reforms and attract FDI to China. We directly extracted the FDI data from China's statistical yearbook.

4. **MOHR**. During the journey of China's reform, the marketization of human resources is generally consistent with the whole market-oriented reforms [32]. We expect to use a regional index of the marketization to gauge the level of marketization of human resources. In recent years, Hu Lipeng, Fan Gang, and Wang Xiaolu, who are from Beijing National Economic Research Institute, have conducted a systematic research on the marketization of China, and released "China's Marketization Index Report by Province (2018)". The index, released from the report, has been generally recognized widely by the Chinese. Therefore, we adopt the index directly.

5. **SLEG**. It is considered that the support from public policies refers to social acceptance for specific matters. Before formulating a policy, China's government generally obtain the views of various sectors of the community. We found that the policies of different provinces are all in favor of the HRS. There are also differences between the time of the policies released. Moreover, it is well known that the earlier the policy is formulated, the wider the social recognition of the HRS in China. Therefore, we calculated the interval between the initialing time of the policy and the research data statistical deadline. We believe that the interval could measure the different social legitimacy between the provinces. Additionally, we calculated 1 month simply because the interval is not a full month.

6. **HRSI.** There is a systematic research report that was jointly released by the Human Resources Development and Management Research Center of Peking University and Shanghai Foreign Service (Group) Co., Ltd. Many indicators from the report have been widely acknowledged by Chinese scholars and local governments. Consequently, we believe that the development index of HRSI from the report reflects the sustainable development of the regional HRSI.

Finally, this study carried out a lagging design for the data collection process. The time span of the study was from the end of 2017 to the end of 2019. We collected data of GDP, PLR, FDI, and MOHR at the end of 2017, and got data about HRSI published in 2020 (the data statistics deadline is the end of 2019). In addition, we acquired the LEGI indicator in late 2018. We believe that the indicator of 2018 could be applied to 2017 because of governments' acquiescence for the HRS before a special policy is issued generally. Moreover, we consider that such treatment of the SLEG indicator has no significant impact on our results. The detailed data sources of the five conditions and outcome are presented in the Table A1 in Appendix A.

### 3.3. Initial Calibration

The calibration consists of converting the variables into fuzzy variables, assigning them values between 0.0 and 1.0 according to their degree of membership or integration [40]. As preparations for fsQCA analysis, we need to pre-calibrate the conditions and outcome. This pre-calibration reckons the considered conditions and outcome, originally in their own scales, to fuzzy membership scores ranging from 0 to 1 [41]. And, the raw data of

these conditions are presented in the Table A2 in Appendix A. To reduce the redundancy of data, we standardized the data about GDP, PLR, and FDI before the calibration.

During the calibration process, this context followed the guidance of the method described by Ragin (2008). And, more recently, there is a developed version outlined by Andrews, Beynon, and McDermott [42]. The version employed probability density functions (pdfs) constructed for conditions and an outcome. The requirement of the techniques is establishing three qualitative thresholds for each condition and the outcome. And then, the thresholds are used in the log-odds transform to calculate the concomitant degrees of membership values [12]. These thresholds refer to the upper threshold—x3 (95th percentile of constructed pdf), crossover—x2 (50th percentile of constructed pdf), and lower threshold—x1(5th percentile of constructed pdf) in this research.

However, considering that the outcome has been clearly divided into 4 categories with cutting points (−1, 0 and 1) by the authors of the report, this paper uses −1, 0, and 1 as the outcome thresholds. Moreover, we did not change the anchors of various conditions mentioned above. Consequently, we checked the established thresholds following the process used by Andrews, Beynon, and McDermott [42]. Finally, the antecedents and the outcome comprised the following variables and cutoff points for continuous variables (the detail is present in Table 1).

**Table 1.** Calibration of the outcome and five conditions.

| Variable | Descriptor | Full Membership, Crossover Point, Full Non Membership |
|---|---|---|
| GDP | Regional gross domestic product | x3, x2, x1 |
| PLR | Regional population of long-term residents | x3, x2, x1 |
| FDI | Regional foreign direct investment | x3, x2, x1 |
| MOHR | Regional marketization index of China | x3, x2, x1 |
| SLEG | Number of months from issuance of regional key policies to statistical deadline | x3, x2, x1 |
| HRSI | the development level of HRS industry in China | 1, 0, −1 |

## 4. fsQCA Analysis

According to the method mentioned above, we produced a data matrix (the data is presented in Table 2). All the data was produced following the method mentioned above. For example, the GDP of Beijing is 30319.98 (the unit of measurement is 100 million yuan). Firstly, we standardized the data. The outcome of Ln (30319.98) is 10.31956. Secondly, we calculated the quartiles of all the Ln (GDP). The quartiles are 9.661731, 9.998106, and 10.497296, respectively. Thirdly, we computed the thresholds for all Ln (GDP). The value of the thresholds is 8.089046 (full non membership), 9.998206 (crossover point) and 11.340422 (Full membership) respectively. Finally, we calibrated the fuzzy value of 10.31956 among the whole Ln (GDP). The value of GDP is just 0.669287.

The data matrix demonstrates that although China's provinces all focus on the growth of the HRSI, there has been a significant gap between developed regions and underdeveloped areas, such as Shanghai and Tibet. This gap indicates the complexity of causality between lots of factors and the development of HRS. The resembled public policies do not necessarily produce similar results. Explaining the gap should help us to improve the effectiveness of public policies to promote the growth of the HRSI.

Then, we followed the iterative approach of fsQCA analysis, which consisted of necessity and sufficiency analyses. Both analyses attempt to explain the consistency of the direction of the condition's change with the result's change. Moreover, the analyses indicate the extent to which the sample is covered by a particular theoretical explanation. The calculation process has been realized automatically by R software. The analyses are described in the following sections.

**Table 2.** Data matrix.

| NO. | Cases | GDP | PLR | FDI | MOHR | SLEG | HRSI |
|---|---|---|---|---|---|---|---|
| 1 | Beijing | 0.669287 | 0.281875 | 0.899653 | 0.903256 | 0.773266 | 0.99536 |
| 2 | Tianjin | 0.440109 | 0.185257 | 0.797613 | 0.94439 | 0.876852 | 0.981947 |
| 3 | Hebei | 0.746925 | 0.89258 | 0.531924 | 0.410639 | 0.773266 | 0.333461 |
| 4 | Shanxi | 0.398117 | 0.48546 | 0.318246 | 0.232685 | 0.561035 | 0.159954 |
| 5 | Inner Mongolia | 0.408369 | 0.335259 | 0.295409 | 0.105783 | 0.05 | 0.046416 |
| 6 | Liaoning | 0.576697 | 0.600927 | 0.838134 | 0.499973 | 0.940498 | 0.361284 |
| 7 | Jilin | 0.358447 | 0.362115 | 0.249825 | 0.486294 | 0.474636 | 0.2708 |
| 8 | Heilongjiang | 0.387992 | 0.494924 | 0.214715 | 0.338988 | 0.05 | 0.039224 |
| 9 | Shanghai | 0.704618 | 0.319138 | 0.943915 | 0.951311 | 0.991193 | 1 |
| 10 | Jiangsu | 0.95907 | 0.910821 | 0.955445 | 0.912625 | 0.958052 | 0.999702 |
| 11 | Zhejiang | 0.886815 | 0.772433 | 0.864978 | 0.953014 | 0.889499 | 0.999403 |
| 12 | Anhui | 0.664227 | 0.823187 | 0.499966 | 0.578768 | 0.620279 | 0.418038 |
| 13 | Fujian | 0.744536 | 0.515258 | 0.802268 | 0.90407 | 0.590987 | 0.521567 |
| 14 | Jiangxi | 0.499961 | 0.642572 | 0.476036 | 0.567334 | 0.05 | 0.140528 |
| 15 | Shandong | 0.939023 | 0.95329 | 0.831557 | 0.752548 | 0.86298 | 0.959487 |
| 16 | Henan | 0.847522 | 0.94646 | 0.55937 | 0.581045 | 0.561035 | 0.794067 |
| 17 | Hubei | 0.782069 | 0.795022 | 0.589285 | 0.662157 | 0.375739 | 0.962325 |
| 18 | Hunan | 0.751652 | 0.861607 | 0.691345 | 0.574204 | 0.590987 | 0.315163 |
| 19 | Guangdong | 0.963113 | 0.966492 | 0.978752 | 0.948647 | 0.05 | 0.999855 |
| 20 | Guangxi | 0.470252 | 0.681475 | 0.355738 | 0.413291 | 0.05 | 0.016733 |
| 21 | Hainan | 0.08812 | 0.088268 | 0.455545 | 0.166706 | 0.05 | 0.024076 |
| 22 | Chongqing | 0.470453 | 0.41022 | 0.527733 | 0.78727 | 0.073215 | 0.757242 |
| 23 | Sichuan | 0.794074 | 0.918994 | 0.583045 | 0.576488 | 0.05 | 0.372331 |
| 24 | Guizhou | 0.352106 | 0.471687 | 0.197985 | 0.111073 | 0.05 | 0.070933 |
| 25 | Yunnan | 0.42097 | 0.669444 | 0.239904 | 0.082549 | 0.05 | 0.128964 |
| 26 | Tibet | 0.015305 | 0.021655 | 0.009959 | 0.047956 | 0.05 | 0.007029 |
| 27 | Shaanxi | 0.557712 | 0.499959 | 0.472813 | 0.450875 | 0.499997 | 0.144759 |
| 28 | Gansu | 0.180565 | 0.349192 | 0.119443 | 0.081724 | 0.05 | 0.010096 |
| 29 | Qinghai | 0.041374 | 0.045141 | 0.034869 | 0.024131 | 0.676143 | 0.037356 |
| 30 | Ningxia | 0.060293 | 0.055335 | 0.192179 | 0.146491 | 0.05 | 0.010781 |
| 31 | Xinjiang | 0.2873 | 0.323017 | 0.071195 | 0.052116 | 0.620279 | 0.01521 |

Cases: province of China mainland. GDP: the membership of gross domestic product. PLR: the membership of population of long-term residents. FDI: the membership of foreign direct invest. MOHR: the membership of marketization index of human resource. SLEG: the membership of index of social legitimacy. HRSI: the membership of human resources services industry development level.

### 4.1. Necessity Analysis

The necessity analysis in fsQCA examines whether an individual condition may be necessary for the outcome to occur. For necessary conditions, the membership score on the outcome must be consistently lower than the membership score of the conditions under consideration [43]. Given the asymmetry of fsQCA [12], results for the two outcomes (HRSI and ~HRSI) are described in Table 3.

Conventionally, a condition is called "necessary" or "almost always necessary" if the consistency score exceeds 0.9 [44–46]. We analyze whether any of the five causal conditions is necessary for a high or low sustainable development level of the HRSI. The consistency measure of the MOHR is 0.911, which is above 0.9 (the 90 percent of the consistency threshold). Furthermore, the values of the RoN and the covN are 0.818 and 0.771, respectively (see Table 3). The results tell us that the MOHR can be considered a necessary condition for the boom of the services. Overall, our findings are consistent with hypothesis 4.

**Table 3.** Analysis of necessity for human resources services industry (HRSI and ~HRSI).

| Conditions | | Outcome–HRSI | | | | | |
|---|---|---|---|---|---|---|---|
| | | **HRSI** | | | **~HRSI** | | |
| | | inclN | RoN | covN | inclN | RoN | covN |
| GDP | Var | 0.857 | 0.729 | 0.671 | 0.480 | 0.652 | 0.528 |
| | ~Var | 0.397 | 0.636 | 0.352 | 0.701 | 0.899 | 0.873 |
| PLR | Var | 0.762 | 0.676 | 0.589 | 0.537 | 0.673 | 0.583 |
| | ~Var | 0.460 | 0.665 | 0.414 | 0.622 | 0.845 | 0.786 |
| FDI | Var | 0.882 | 0.785 | 0.729 | 0.424 | 0.660 | 0.492 |
| | ~Var | 0.385 | 0.559 | 0.332 | 0.766 | 0.911 | 0.901 |
| MOHR | Var | 0.911 | 0.818 | 0.771 | 0.385 | 0.656 | 0.485 |
| | ~Var | 0.359 | 0.578 | 0.293 | 0.807 | 0.930 | 0.927 |
| SLEG | Var | 0.725 | 0.817 | 0.702 | 0.371 | 0.729 | 0.505 |
| | ~Var | 0.489 | 0.539 | 0.356 | 0.781 | 0.789 | 0.799 |

Notes: "Var" indicates high level of the special condition, while "~ Var" provides low level of the special condition. "HRSI" refers to high sustainable development degree of HRSI, "~HRSI" stands for low growth degree of HRSI. inclN: necessity inclusion score (the proportion of outcome in the intersection between condition X and an outcome). RoN: relative importance of that condition as a necessary condition. CovN: raw necessity coverage (the relevance of a necessary condition X for an outcome).

### 4.2. Sufficiency Analysis

After analyzing the necessary conditions, we took a sufficiency analysis to identify conditions that lead to HRSI or ~HRSI. In the analysis, we found different combinations of conditions that meet specific criteria of sufficiency for the outcome to exist.

#### 4.2.1. Truth Tables

The precondition for sufficiency analysis is the truth tables, which list all logically possible combinations, termed configurations, and the outcome (either HRSI or ~HRSI).

With conditions analyzed in our research, there are $2^5 = 32$ logically possible configurations that should be considered. The configurations are marked by "0" (standing for the absence of the condition) and "1" (implying the presence of the condition) across the five conditions. The value of each outcome is "0" (referring to the low development level of HRSI) or "1" (meaning high growth of HRSI) following the same logic of the calibration.

For each case observation, the outcome or a condition's membership score (either "0" or "1") is based on a strong membership. The threshold values are the same for HRSI or ~HRSI in this paper. Furthermore, the level of consistency shows that the membership in the outcome is consistently less than or equal to the membership in the cause [12]. A consistency value above 0.78 is used to ensure that no configuration has an association with both HRSI and ~HRSI in order to avoid a logic contradiction. Moreover, the frequency is the minimum number of province observations that should be further analyzed. Considering that there are not too many samples, this research sets the frequency to 1. The details are presented in Tables 4 and 5, which refers to the high and low development degree of HRSI, respectively.

**Table 4.** Truth table regarding HRSI.

| NO. | GDP | PLR | FDI | MOHR | SLEG | OUT | n | Incl | PRI | Cases |
|---|---|---|---|---|---|---|---|---|---|---|
| 24 | 1 | 0 | 1 | 1 | 1 | 1 | 2 | 0.886 | 0.790 | Beijing, Shanghai |
| 32 | 1 | 1 | 1 | 1 | 1 | 1 | 6 | 0.864 | 0.777 | Jiangsu, Zhejiang, Fujian, Shandong, Henan, Hunan |
| 8 | 0 | 0 | 1 | 1 | 1 | 1 | 1 | 0.862 | 0.717 | Tianjin |
| 28 | 1 | 1 | 0 | 1 | 1 | 0 | 1 | 0.829 | 0.597 | Anhui |
| 30 | 1 | 1 | 1 | 0 | 1 | 0 | 2 | 0.774 | 0.485 | Hebei, Liaoning |
| 31 | 1 | 1 | 1 | 1 | 0 | 0 | 3 | 0.760 | 0.574 | Hubei, Guangdong, Sichuan |
| 7 | 0 | 0 | 1 | 1 | 0 | 0 | 1 | 0.676 | 0.386 | Chongqing |
| 11 | 0 | 1 | 0 | 1 | 0 | 0 | 1 | 0.660 | 0.288 | Jiangxi |
| 2 | 0 | 0 | 0 | 0 | 1 | 0 | 3 | 0.506 | 0.167 | Shanxi, Qinghai, Xinjiang |
| 17 | 1 | 0 | 0 | 0 | 0 | 0 | 1 | 0.465 | 0.124 | Shaanxi |
| 9 | 0 | 1 | 0 | 0 | 0 | 0 | 2 | 0.455 | 0.104 | Guangxi, Yunnan |
| 1 | 0 | 0 | 0 | 0 | 0 | 0 | 8 | 0.281 | 0.060 | Inner Mongolia, Jilin, Heilongjiang, Hainan, Guizhou, Tibet, Gansu, Ningxia |

NO.: observable configurations of conditions. GDP: the condition of high gross domestic product. PLR: the condition of large population of long-term residents. FDI: the condition of large-scale foreign direct investment. MOHR: the condition of high index of marketization of human resource. SLEG: the condition of sufficient social legitimacy. OUT: high growth rate of the HRSI. n: number of cases in particular combinations of conditions. incl: sufficiency inclusion score. PRI: proportional reduction in inconsistency. cases: provinces of China in this context.

**Table 5.** Truth table regarding ~HRSI.

| NO. | GDP | PLR | FDI | MOHR | SLEG | OUT | n | Incl | PRI | Cases |
|---|---|---|---|---|---|---|---|---|---|---|
| 1 | 0 | 0 | 0 | 0 | 0 | 1 | 8 | 0.954 | 0.940 | Inner Mongolia, Jilin, Heilongjiang, Hainan, Guizhou, Tibet, Gansu, Ningxia |
| 9 | 0 | 1 | 0 | 0 | 0 | 1 | 2 | 0.937 | 0.896 | Guangxi, Yunnan |
| 17 | 1 | 0 | 0 | 0 | 0 | 1 | 1 | 0.924 | 0.876 | Shaanxi |
| 2 | 0 | 0 | 0 | 0 | 1 | 1 | 3 | 0.901 | 0.833 | Shanxi, Qinghai, Xinjiang |
| 11 | 0 | 1 | 0 | 1 | 0 | 1 | 1 | 0.862 | 0.712 | Jiangxi |
| 7 | 0 | 0 | 1 | 1 | 0 | 1 | 1 | 0.796 | 0.614 | Chongqing |
| 30 | 1 | 1 | 1 | 0 | 1 | 1 | 2 | 0.788 | 0.515 | Hebei, Liaoning |
| 28 | 1 | 1 | 0 | 1 | 1 | 0 | 1 | 0.747 | 0.403 | Anhui |
| 31 | 1 | 1 | 1 | 1 | 0 | 0 | 3 | 0.677 | 0.426 | Hubei, Guangdong, Sichuan |
| 8 | 0 | 0 | 1 | 1 | 1 | 0 | 1 | 0.649 | 0.283 | Tianjin |
| 24 | 1 | 0 | 1 | 1 | 1 | 0 | 2 | 0.570 | 0.210 | Beijing, Shanghai |
| 32 | 1 | 1 | 1 | 1 | 1 | 0 | 6 | 0.526 | 0.223 | Jiangsu, Zhejiang, Fujian, Shandong, Henan, Hunan |

NO.: observable configurations of conditions. GDP: the condition of high gross domestic product. PLR: the condition of large population of long-term residents. FDI: the condition of large-scale foreign direct investment. MOHR: the condition of high index of marketization of human re-source. SLEG: the condition of sufficient social legitimacy. OUT: low sustainable development level of the HRSI. n: number of cases in particular combinations of conditions. incl: sufficiency inclusion score. PRI: proportional reduction in inconsistency. cases: provinces of China in this context.

4.2.2. Sufficiency Analysis of High Development Degree of HRSI

We firstly analyzed the sufficient conditions for the presence of a high sustainable development degree of HRSI and then examined the low sustainable development level of HRSI. There are three types of solutions: complex, parsimonious and intermediate solutions. The complex solution did not limit logical configurations, while the latter two types of solutions addressed the limited remainders [12].

Moreover, the latter two dealt with limited diversity in different ways, with the parsimonious solution using each logical remainder, and the intermediate solution only relying on "easy" counterfactuals. "Easy" counterfactuals stand for situations in which a redundant causal condition is added to a set of causal conditions that by themselves already lead to the outcome (i.e., high development level of HRSI) in question. In contrast, the "difficult" counterfactuals refer to situations in which a condition is removed from a set of causal conditions leading to an outcome assuming that this condition is redundant [47].

We focused on the conservative solution formula because the parsimonious solution is unrealistically simple, and the intermediate solution may make the distinction between theory and the empirical analysis unclear [48].

The complex solution formulas to the high development level of HRSI is GDP*PLR* MOHR*SLEG + ~PLR*FDI*MOHR*SLEG → HRSI, and the parsimonious solution formula is MOHR*SLEG → HRSI. The complex solution formula, provided in Table 6, includes two paths. While, the parsimonious solution formula, provided in Table A3 in Appendix A, includes only one way, i.e., only one configuration (high MOHR and more SLEG occur at the same time) that is sufficient for high growth level of HRSI.

**Table 6.** Complex solutions in explaining high growth level of HRSI.

|   |   | inclS | PRI | covS | covU | Cases |
|---|---|---|---|---|---|---|
| 1 | GDP*PLR*MOHR*SLEG | 0.883 | 0.828 | 0.698 | 0.142 | Anhui, Jiangsu, Zhejiang, Fujian, Shandong, Henan, Hunan |
| 2 | ~PLR*FDI*MOHR*SLEG | 0.847 | 0.752 | 0.558 | 0.002 | Tianjin, Beijing, Shanghai |
|   | M1 | 0.869 | 0.804 | 0.667 |   |   |

M1: GDP*PLR*MOHR*SLEG + ~PLR*FDI*MOHR*SLEG -> HRSI. inclS: sufficiency inclusion score. PRI: proportional reduction in inconsistency. covS: raw sufficiency coverage (the relevance of a sufficient condition X for an outcome). covU: unique sufficiency coverage (the unique relevance of a sufficient condition X for an outcome). cases: provinces of China.

**Path 1.** *The combination of a high gross domestic product (GDP), a large population of long-term residents (PLR), a high index of marketization of human resources (MOHR), and sufficient social legitimacy (SLEG) results in a high sustainable HRSI development level.*

This path covers seven cases: Anhui, Jiangsu, Zhejiang, Fujian, Shandong, Henan, and Hunan. There are strong economies and lots of jobs in these cases. Furthermore, the governments of these cases pay more attention to the cultivation of the HRSI.

Among them, Shandong and Henan are the most populous provinces with approximately 100 million long-term residents. At the same time, Jiangsu and Zhejiang have been the top economically developed provinces. There is a prosperous economy and high level of marketization of human resources in these two provinces, attracting large number of residents from other regions for employment. Rapid economic development and a large number of employable populations drive the development of the HRSI. Moreover, these seven cases have discovered the importance of HRSI and comprehensively sped up market-oriented market reform and supported the development of the HRSI earlier than other cases, thus leading the HRSI development for a long time.

**Path 2.** *The combination of a small population of long-term residents (~PLR), large-scale foreign direct investment (FDI), a high index of marketization of human resources (MOHR), and sufficient social legitimacy (SLEG) leads to a high sustainable HRSI development level.*

This path covers three cases: Tianjin, Beijing, and Shanghai. These cases are all municipalities directly under the central government in China. Among them, Shanghai

is the economic center, and Beijing is the political and cultural center of China. Both Beijing and Shanghai are locations of the corporate headquarters for most enterprise in China. Meanwhile, Tianjin is not only close to Beijing, but also enjoys very convenient maritime transportation and cross-international exchange conditions. Recently, Tianjin has become the largest open coastal city in north China and the center of the Bohai Rim Economic Circle.

Although the three cases are limited by geographical space and have finite long-term residents, their urbanization level exceeds 80%, making them the most modernized provinces in China. According to official statistics from the Chinese government, the annual income of above the three cases has exceeded RMB 100,000 (approximately 16055.5 USD) per capita in 2015. The degree of openness and the level of marketization in these cases is much higher than most of the other provinces in China. Therefore, their focus on marketization, foreign investment, and higher quality of economic development with early support for HRSI, effectively compensates for the disadvantages of the employable population and the general GDP.

### 4.2.3. Sufficiency Analysis of Low Development Level of HRSI

We proceeded to analyze sufficient conditions that lead to the low sustainable development level of HRSI following the same procedures as showed in the previous section. Table 7 reports that five paths can lead to a low development level of HRSI. The parsimonious solution formula is ~MOHR + ~GDP*~SLEG →~HRSI, the detail of the parsimonious solutions is presented in Table A4 in Appendix A.

**Table 7.** Complex solutions in explaining the low development level of HRSI.

| | | inclS | PRI | covS | covU | Cases |
|---|---|---|---|---|---|---|
| 1 | ~GDP*~PLR*~FDI*~MOHR | 0.955 | 0.943 | 0.542 | 0.056 | Inner Mongolia, Jilin, Heilongjiang, Hainan, Guizhou, Tibet, Gansu, Ningxia, Shanxi, Qinghai, Xinjiang |
| 2 | ~GDP*PLR*~FDI* ~SLEG | 0.901 | 0.839 | 0.376 | 0.040 | Guangxi, Yunnan; Jiangxi |
| 3 | ~PLR*~FDI* ~MOHR*~SLEG | 0.955 | 0.941 | 0.488 | 0.006 | Inner Mongolia, Jilin, Heilongjiang, Hainan, Guizhou, Tibet, Gansu, Ningxia, Shaanxi |
| 4 | ~GDP*~PLR*FDI*MOHR*~SLEG | 0.796 | 0.614 | 0.211 | 0.003 | Chongqing |
| 5 | GDP*PLR*FDI* ~MOHR*SLEG | 0.788 | 0.515 | 0.222 | 0.053 | Hebei, Liaoning |
| | M1 | 0.883 | 0.843 | 0.680 | | |

M1: ~GDP*~PLR*~FDI*~MOHR + ~GDP*PLR*~FDI*~SLEG + ~PLR*~FDI*~MOHR*~SLEG + ~GDP*~PLR*FDI*MOHR*~SLEG + GDP*PLR*FDI*~MOHR*SLEG -> ~HRSI. inclS: sufficiency inclusion score. PRI: proportional reduction in inconsistency. covS: raw sufficiency coverage (the relevance of a sufficient condition X for an outcome). covU: unique sufficiency coverage (the unique relevance of a sufficient condition X for an outcome).

**Path 3.** *The combination of factors due to the low gross domestic product (~GDP), small population of long-term residents (~PLR), little foreign direct investment (~FDI), and low index of marketization of human resources (~MOHR) brings about low sustainable development level of regional HRSI.*

This path covers eleven cases: Inner Mongolia, Jilin, Heilongjiang, Hainan, Guizhou, Tibet, Gansu, Ningxia, Shanxi, Qinghai, and Xinjiang. The private economy in those provinces is very weak, and there are multiple difficulties for the transformation of the state-owned economy. Because of little private economy to create new jobs, a lot of human resources flow out from those regions. In addition, because they are far from the core areas of China's economic development, with fragile natural conditions and long-term resident population disadvantage, these cases have been the most underdeveloped areas for a long time. The first aim of these provinces is to solve the problem of survival, while development issues such as accelerating market reforms and attracting FDI came in second

place. China's government has made outstanding achievements in poverty alleviation and the development of poverty-stricken areas, but the measures adopted to solve poverty by the central government, such as financial assistance, have modestly improved the living conditions of residents in the short term. However, there has been severe insufficiency and imbalance of development in these areas for a long time. Unsurprisingly, human resources development and forging investment are both inadequate for regional development, which intensifies the long-term lag of HRSI sustainable growth in these cases.

**Path 4.** *The combination of low gross domestic product (~GDP), big population of long-term residents (PLR), insufficient foreign direct investment (~FDI), and little social legitimacy (~SLEG) gives rise to little sustainable development of HRSI.*

This path covers three cases: Guangxi, Yunnan and Jiangxi. The common advantage of these three cases is that there is a large employable population, while the employable population is distributed mainly in the agricultural field. According to the statistics of GDP in China, the output of the agricultural sector is lower than that of the other industrial sectors due to the long-term impact of the price scissors gap between industry and agriculture. The GDP of areas dominated by the agriculture sector is lower than that of regions dominated by industry and services. Finally, the share of industry and services in less developed provinces are much lower than that of developed provinces. Among the three cases, Jiangxi did not grasp new opportunities from national economy reform timely, resulting in economic decline and relatively weak attraction for FDI. Compared to Jiangxi, Guangxi and Yunnan are both located on the Yunnan-Guizhou Plateau. On the plateau, there are lots of difficulties to develop economies arriving from complex geological conditions, fewer capabilities to attract FDI and speed up the development of the HRSI. In addition, there are nearly 30 ethnic minorities in Yunnan, which indicates possible and complex contradictions. The prerequisite for the development of Yunnan is properly handling the complexity between various ethnic groups in order to form a joint developmental force.

**Path 5.** *The combination of factors derived from an inadequate population of long-term residents (~PLR), seldom foreign direct investment (~FDI), low degree of marketization of human resources (~MOHR), and little social legitimacy (~SLEG) generates low sustainable development degree of HRSI.*

This path covers nine cases: Inner Mongolia, Jilin, Heilongjiang, Hainan, Guizhou, Tibet, Gansu, Ningxia and Shaanxi. Among these nine cases, Jilin and Heilongjiang were China's heavy industry bases, with many state-owned enterprises. Of these, Inner Mongolia, Guizhou, Tibet, and Ningxia are ethnic communities. Compared with most of the other provinces with the higher development level of HRSI, there are no sufficient employable residents attributing to few jobs for the employable. Additionally, the lack of marketization concepts is a vital factor leading to underdeveloped economies, low level of MOHR and little FDI for a few years. Moreover, the sustainable development of the HRSI has not been taken seriously in these areas.

**Path 6.** *The combination of low gross domestic product (~GDP), a small population of long-term residents (~PLR), large-scale foreign direct investment (FDI), a high index of marketization of human resources (MOHR), and seldom social legitimacy (~SLEG) causes low sustainable development level of the HRSI.*

This path only covers Chongqing. Chongqing is the only municipality directly under the Central Government in western China. It is located at the junction of the "Belt and Road" and the Yangtze River Economic Belt. In addition, the business environment of Chongqing has been continuously optimized with the implementation of China's western development strategy. FDI in Chongqing continues to maintain a leading position in the western of China. However, there is a small economy compared with other developed provinces. The population of the case is less than 1/5 of Henan. Also, to make matters worse, the agricultural population accounts for a large proportion of the population. Finally,

the demand for HRS is insufficient. Additionally, the development level of industrial enterprises in Chongqing is not very high, and there is rare demand for the HRS with high-quality. Moreover, there is limited support for HRSI from the regional public policies, resulting in a low sustainable development level of the industry.

**Path 7.** *The combination of factors derived from a high gross domestic product (GDP), big populations of regional long-term residents (PLR), high level of foreign direct investment (FDI), low index of marketization of human re-sources (~MOHR), and sufficient HRSI social legitimacy (SLEG) results in low sustainable development degree of HRSI.*

This path covers Hebei and Liaoning. Both cases are dominated by state-owned enterprises, which refers to a very weak private economy. Existing research has found that the awareness of market competition is weak because of state-owned enterprises tending to use bureaucratic systems to allocate resources in the cases. With China's opening to the outside world, although the enterprises are still creating GDP, most of these companies are weak in technological innovation and market competitiveness. Both cases actively attract FDI to help the companies improve innovation abilities and competitiveness. However, there are high costs for the market-oriented reforms, and the reform is full of resistance. Moreover, state-owned enterprises are more cautious in choosing the HRS and have a lower motivation to choose the HRS with high-quality. Finally, the little demand fundamentally leads to a low development level of the HRSI.

## 5. Conclusions and Discussion

In this article, we report a comparative study examining the various factors affecting the development of HRSI across 31 provinces in China. The provinces responded variedly after the central government recommending the development of HRSI. Some provinces paid more attention to HRSI, whereas others acted slowly. This paper has analyzed how the combination of five conditions, namely regional economic volume, size of the regional population of long-term residents, FDI, marketization of human resources, and social legitimacy of HRSI, jointly influence the sustainable development of HRSI in China mainland. However, there is no single sufficient condition explaining the high/low level of development of HRSI.

Firstly, our study shows that the MOHR could be a possible necessary condition for the boom of the HRSI. This finding verifies the simultaneity of employment liberalization and HRS development. The demand for flexible labor embraces the positive role of HRS agencies as human capital managers in the EU [11]. According to a report from the CIETT in 2000, most of the advanced industrial nations have been favorable to the services. To promote the marketization of human resources, the government should adopt effective public policies from the supply side and demand side. On one hand, the government should pay more attention to the training of labors' vocational skills and give assistance to those who have difficulty hunting jobs. On the other hand, the government should encourage employers to hire workers through the human resources market rather than relying on the arrangements of government officials. Moreover, the regional government needs to focus on the governance of the market to guarantee equal employment opportunities and create full employment.

Secondly, there is no single sufficient condition explaining the high/low level of the development of HRSI. We find that a general economy is an important condition for development of HRSI. Eleven of eighteen cases with low development level of HRSI are underdeveloped regions. The eleven provinces occupy more than 90% of western China, where the economic foundation is lacking. Also, there is little attraction for FDI. Consequently, we believe that the major driving force of the industry growth originates from the regional economy. Furthermore, this research demonstrates that a certain economic foundation does not necessarily lead to the development of HRS. In the cases of Hebei and Liaoning, the low level of economic marketization and weak demand of the HRS fundamentally restrict the development of HRS. In other words, the insufficient demand

of the services limits the growth of the HRS [49]. Moreover, even with marketization, FDI and a certain economic foundation, there is still low growth of the HRSI because of little social approval degree for the services. That is to say that the characters of the services require the social acceptance, which is in compliance with social legitimacy research of the services [50].

Thirdly, our study finds two different paths that lead to the high development level of HRSI. Our analysis demonstrates that the configuration of a high level of foreign investment, marketization of human resources, and the social legitimacy of HRS are the core conditions in explaining the sustainable development of the HRSI. Out of the decade cases with the high development level of HRSI, six cases are on the frontline of China's reform and opening, which refers to high foreign investment, marketization of human resources and more social legitimacy of HRS. It seems that the market-oriented reform and the services social legitimacy is a relevant INUS (This means that the condition itself is not sufficient to bring about the outcome of interest. However, this condition is a necessary part of the combination of conditions, which as a whole is sufficient to result in the outcome of interest, but itself is not necessary for the outcome of interest, since there are several paths that could lead to this.) condition in explaining the sustainable development of HRSI. This confirms the sensitivity of HRS to the institutional environment and highlights the important role of multinational HRS providers in promoting the social legitimacy of the services [2]. The governments of economies in transition should has an open and inclusive attitude towards the expansion of the services in host countries.

Meanwhile, there is a coexistence of interacting and counteracting drivers of the growth of the HRS. The economic and human resources foundations are different in each region. Although some regions do not have the advantages of all the influencing factors of HRSI development, the combination of some factors can still fetch up deficiencies of other factors resulting in the development of HRSI. For instance, there are regions with relatively disadvantaged long-term residents and low GDP. The regions could improve the quality of economic development, and guide the development of HRS with high-quality demand, to realize the quality advantage to make up for the quantity disadvantage of the population and the GDP. Moreover, the development of HRSI is not necessarily prosperous for those regions with advantages of the employable and regional economy because that resource advantage has not been transformed into an efficiency advantage. In other words, the resource allocation efficiency by markets is higher than that by non-market-based resource allocation, and the efficiency advantage is as important as quantity for the growth of HRS.

Therefore, we recommend that regional governments adopt systematic and complementary public policies based on regional strengths, weaknesses, opportunities, and threats specific to that area. For instance, it is necessary to consider the natural and human resources endowment of the region systematically and focus on attracting FDI that matches these resources endowment. At the same time, it is urgent to enhance the social status of human resources services providers and encourage the providers to play vital roles in attracting talents, improving the professional skills of workers, and promoting the allocation of human resources. Additionally, policies for attracting FDI should think of the impact of the investment on the development of human resources and regional economic upgrading rather than merely the GDP.

Fourthly, there are five paths that give rise to the low growth degree of the HRSI. The cases following the path either lack of high GDP or high index of MOHR in general. To get rid of the development insufficiency of the services, there are a few ways to promote regional economic development, such as attracting foreign direct investment and exploring available resources efficiently. In order to improve the development quality, the marketization of human resources should be concerned seriously. Lots of researchers have found that the MOHR is a vital part of market-oriented economic reform. The cultivation of the factor market, such as the human resources market, cannot be separated from the support of the construction of the whole market economic system. The reform of the labor system goes hand in hand with the market-oriented reform of the economy. It is necessary to actively

integrate into internationalization through openness and market-oriented reform. On one hand, the regions with the low growth rate of HRSI should accelerate market-oriented reform of regional economy and strengthen the foundation of the economy. On the other hand, the regions should attract FDI fitting the resources of the areas.

Fifthly, this study provides some theoretical insights into the development of HRS in transforming economies. A few studies have revealed the influence of individual factors on the development of HRS, but there is little explanation for the combination of different factors affects the development of HRS. Our research effectively enriches the study of HRS by using the method of QCA. The method explains the complexity of HRS development effectively. It seems that the growth of services in developed countries relies on internationalization and innovation of the services [11,34]. However, transforming economies mainly depends on the demand of the services market. In China, as we have argued, there are a few provinces that strive to speed up the development of the HRSI. Some of the provinces have not achieved the expected results. Considering the paths that lead to different growth rates of HRSI, we find that the best way to cultivate the services is just speeding up market-oriented reform of human resources, promoting economic internationalization, integrating internal and external resources of the region timely, and paying attention to the efficiency and quality of services development.

Last but not the least, we need to emphasize that our study itself does not ambitiously provide a comprehensive explanation of the development of regions in China and other transforming economies. In this paper, we primarily use fsQCA as an exploratory instrument to provide some preliminary clues that should be further investigated by supplementing with other methods such as in-depth case studies, and/or mathematical economic analysis. This means that more follow-up studies are necessary in order to provide a much robust explanation about this issue. Moreover, there are lots of differences between transforming economies, such as different employment cultures, devious roles of labor unions, and different levels of internationalization. Whether the conclusions of this paper about HRS in China can be extended to more transforming economies still needs to be improved through transnational comparative studies.

One practical suggestion for regional governments and HRS companies is that they should promote internationalization and market-oriented reform of the economy to nurture the demand of the HRS with high quality. In regions with fewer human resources advantages, they should increase the social legitimacy of the services based on exploring efficiency of available resources. Only by taking these combined measures can a region enjoy a high development level of the HRS. Moreover, along with the coming of the Fourth Industrial Revolution, the development of digital technology and digital economy will bring about dramatic changes in HRS, such as changes of technology tools, business model, operation of the service, and governance of the services. We believe that although there are experiences which can be learned from the developed countries, transforming countries should seize new opportunities of the revolution based on the cultivation of the services, and participate in internationalization and innovation of the services, to achieve high-quality development of the services and full employment, especially in the post-COVID-19 era.

**Author Contributions:** Conceptualization, W.C. and Y.L.; methodology, W.C.; software, X.-J.S.; validation, W.C. and X.-J.S.; formal analysis, W.C.; resources, W.C.; data curation, X.-J.S.; writing—original draft preparation, W.C.; writing—review and editing, X.-J.S.; supervision, Y.L.; funding acquisition, X.-J.S. and Y.L. All authors have read and agreed to the published version of the manuscript.

**Funding:** This research was supported by Qing Lan Project of Higher Education in Jiangsu Province (2019), Fundamental Research Fund for the Central Universities (LGZD201807), Pre-research Fund of Nanjing Forest Police College (LGY201701), Natural Science Foundation of Jiangsu Province (BK20181338), Excellent Scientific and Technological Innovation Team of Higher Education in Jiangsu Province (2019-29) and Chinese National Funding of Social Sciences (15ZDC014).

**Institutional Review Board Statement:** Not applicable.

**Informed Consent Statement:** Not applicable.

**Data Availability Statement:** The input data in this research can be accessed freely from online sources. The detail of the data source was mentioned in Table A1.

**Acknowledgments:** The authors are grateful to Yanwei Li in School of Public Administration from Nanjing Normal University China and Huang Jin in School of Government from Nanjing University China for their excellent research method assistance. The authors would like to thank anonymous reviewers and the editor for helpful comments and suggestions.

**Conflicts of Interest:** The authors declare no conflict of interest.

## Appendix A

**Table A1.** Data source of five conditions and the outcome.

| Conditions and Outcome | Data Source |
|---|---|
| Regional gross domestic product (GDP) | Gross regional product (2018) from China Statistical Yearbook (2018) |
| Population of long-term resident (PLR) | Statistical Communiqués issued by each Provincial Statistics Bureau of China |
| Foreign direct investment (FDI) | Number and Investment of Registered Enterprise with Foreign Capital by Region or Department at the Year 2017-end from China Statistical Yearbook (2018) |
| Marketization of human resources (MOHR) | "China's Marketization Index Report by Province (2018)" published by Social Science Literature Press in February 2019 |
| Social legitimacy (SLEG) | Websites of each Provincial Department of Human Resources and Social Security in China |
| HRS industry (HRSI) | "Human resources Services Industry Development Level Ranking List of China's Provinces and Urban Areas (2020)" issued by the Human resources Development and Management Research Center of Peking University and Shanghai Foreign Service (Group) Co., Ltd. |

Data source: The authors' collation.

**Table A2.** Raw data of the five conditions and outcome.

| Province | Quantity of GDP (Statistical Unit: 100 Million Yuan) | Population of Long-Term Resident (Statistical Unit: 10 Thousands) | Quantity of Foreign Direct Investment (Statistical Unit: 10 Thousands US Dollars) | Index of Marketization | Social Legitimated Months | Index of HRSI Development |
|---|---|---|---|---|---|---|
| Beijing | 30,319.98 | 2170.7 | 48,640,860 | 9.14 | 50 | 1.823246 |
| Tianjin | 18,809.64 | 1556.87 | 25,482,286 | 9.78 | 62 | 1.357213 |
| Hebei | 36,010.27 | 7519.52 | 9,581,812 | 6.42 | 50 | −0.23521 |
| Shanxi | 16,818.11 | 3702.35 | 4,972,449 | 5.66 | 34 | −0.56329 |
| Inner Mongolia | 17,289.22 | 2528.6 | 4,597,937 | 4.8 | 1 | −1.02654 |
| Liaoning | 25,315.35 | 4368.9 | 31,585,001 | 6.75 | 75 | −0.19352 |
| Jilin | 15,074.62 | 2717.43 | 3,887,356 | 6.7 | 29 | −0.33642 |
| Heilongjiang | 16,361.62 | 3788.7 | 3,366,862 | 6.14 | 1 | −1.08627 |
| Shanghai | 32,679.87 | 2418.33 | 79,823,905 | 9.93 | 107 | 4.015547 |
| Jiangsu | 92,595.4 | 8029.3 | 96,581,873 | 9.26 | 81 | 2.756553 |
| Zhejiang | 56,197.15 | 5657 | 37,341,457 | 9.97 | 64 | 2.521131 |
| Anhui | 30,006.82 | 6254.8 | 8,664,121 | 7.09 | 38 | −0.11236 |
| Fujian | 35,804.04 | 3911 | 26,072,064 | 9.15 | 36 | 0.029317 |
| Jiangxi | 21,984.78 | 4622.1 | 8,079,723 | 7.04 | 1 | −0.61503 |
| Shandong | 76,469.67 | 10,005.83 | 30,421,775 | 7.94 | 60 | 1.074832 |
| Henan | 48,055.86 | 9559.13 | 10,453,774 | 7.1 | 34 | 0.458362 |
| Hubei | 39,366.55 | 5902 | 11,510,270 | 7.47 | 25 | 1.100501 |
| Hunan | 36,425.78 | 6860.2 | 16,339,193 | 7.07 | 36 | −0.26358 |
| Guangdong | 97,277.77 | 11169 | 176,222,731 | 9.87 | 1 | 3.00126 |
| Guangxi | 20,352.51 | 4885 | 5,620,020 | 6.43 | 1 | −1.38345 |
| Hainan | 4832.05 | 925.76 | 7,608,902 | 5.28 | 1 | −1.25734 |
| Chongqing | 20,363.19 | 3075.16 | 9,455,839 | 8.15 | 5 | 0.386362 |
| Sichuan | 40,678.13 | 8302 | 11,279,723 | 7.08 | 1 | −0.17736 |
| Guizhou | 14,806.45 | 3580 | 3,125,132 | 4.85 | 1 | −0.87366 |
| Yunnan | 17,881.12 | 4800.5 | 3,738,226 | 4.55 | 1 | −0.64873 |
| Tibet | 1477.63 | 377 | 303,136 | 4.02 | 1 | −1.68137 |
| Shaanxi | 24,438.32 | 3835.44 | 8,003,950 | 6.57 | 30 | −0.60328 |
| Gansu | 8246.07 | 2625.71 | 2,019,750 | 4.54 | 1 | −1.55732 |
| Qinghai | 2865.23 | 598.38 | 769,932 | 3.37 | 42 | −1.1035 |
| Ningxia | 3705.18 | 681.79 | 3,042,021 | 5.14 | 1 | −1.53479 |
| Xinjiang | 12,199.08 | 2444.67 | 1,332,275 | 4.1 | 38 | −1.41639 |

Data source: the authors' collation.

**Table A3.** The parsimonious solutions in explaining high development level of HRSI.

|  |  | inclS | PRI | covS | covU | Cases |
|---|---|---|---|---|---|---|
| 1 | MOHR * SLEG | 0.857 | 0.795 | 0.708 | - | Tianjin, Beijing, Shanghai, Anhui, Jiangsu, Zhejiang, Fujian, Shandong, Henan, Hunan |
|  | M1 | 0.857 | 0.795 | 0.708 |  |  |

M1: MOHR * SLEG -> HRSI. incl: sufficiency inclusion score. PRI: proportional reduction in inconsistency. covS: raw sufficiency coverage (the relevance of a sufficient condition X for an outcome). covU: unique sufficiency coverage (the unique relevance of a sufficient condition X for an outcome).

**Table A4.** Parsimonious solutions in explaining low development level of HRSI.

|  |  | inclS | PRI | covS | covU | Cases |
|---|---|---|---|---|---|---|
| 1 | ~MOHR | 0.927 | 0.906 | 0.807 | 0.222 | Inner Mongolia, Jilin, Heilongjiang, Hainan, Guizhou, Tibet, Gansu, Ningxia, Shanxi, Qinghai, Xinjiang, Guangxi, Yunnan, Shaanxi, Hebei, Liaoning |
| 2 | ~GDP *~SLEG | 0.916 | 0.889 | 0.609 | 0.024 | Inner Mongolia, Jilin, Heilongjiang, Hainan, Guizhou, Tibet, Gansu, Ningxia; Chongqing; Guangxi, Yunnan; Jiangxi |
|  | M1 | 0.899 | 0.870 | 0.831 |  |  |

From C1P1: M1: ~MOHR + ~GDP *~SLEG -> ~HRSI. incl: sufficiency inclusion score. PRI: proportional reduction in inconsistency. covS: raw sufficiency coverage (the relevance of a sufficient condition X for an outcome). covU: unique sufficiency coverage (the unique relevance of a sufficient condition X for an outcome).

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
