# Peer review of "Factors Affecting the Sustainable Development of HRS in Transforming Economies: A fsQCA Approach"

_sustainability, doi:10.3390/su13041727_

Round 1
Reviewer 1 Report
Line 71, why Labor should be capital? The authors need to make sure where capital must be used or not. Other than that, I enjoyed reading the paper.
All the statistics that the authors used in the paper are amazing and they did a great job to put it in a simple way when the whole statistics and data analysis process is not simple.
I think that this manuscript is good to go as it is, but only one comment would be just check all the grammars, punctuations and also basic English writing rules.
Author Response
Dear reviewer,
Thank you very much for your message concerning our manuscript sustainability-1069969. We genuinely appreciate your valuable comments and suggestions. According to the suggestions, the manuscript has been revised carefully.
Comment 1: Line 71, why Labor should be capital? The authors need to make sure where capital must be used or not. Other than that, I enjoyed reading the paper.
Response: This question is marvelous for our modification of this article. We have checked the “capital” used in the context and rewrote the sentences in row 94-95 in the revision.
Comment 2: All the statistics that the authors used in the paper are amazing and they did a great job to put it in a simple way when the whole statistics and data analysis process is not simple.
Response: Thank you for your recognition. This study is just a part of our research, and we will continue to work on it.
Comment 3: I think that this manuscript is good to go as it is, but only one comment would be just check: all the grammars, punctuations and also basic English writing rules.
Response: Thank you for your valuable suggestion. English language has been checked carefully and necessary changes were made in the revision.
Moreover, we have followed other reviewers’ suggestions and modified the context.
Reviewer 2 Report
REVIEW
Sustainability
Manuscript #: 1069969
Title: Factors Affecting the Sustainable Development of HRS in Transforming Economies: A fsQCA Approach
Summary: The study uses a fsQCA approach, merging qualitative and quantitative methodologies, to explore configurations of factors which lead (or not) to the development of human resource service. The data used in the study came from Chinese provinces.
Impression: This is an interesting study with a rarely seen methodology. On the other hand, it leaves many terms undefined, its policy implications and conclusion are weak, and above all, the English language in many parts of the manuscript is hard if not impossible to understand. All three facets need some serious extra work.
Row #:
- Human Resource Service (HRS) was not defined and an internet search wasn’t much help either. Since HRS is in the heart of the study, it is critical to explain the concept to the reader in detail.
- What is “marketization of human resources” (MOHR)?
22-23. “[…] transforming economies should […] give impetus to economy transformation.” I don’t think anyone argues against this. Delete.
- Define Human Resource Service clearly.
35-36. “[…] promoted the regional employment, but also created numerous jobs.” How else?
75-76. “[…] the more regional employable population, the larger sustainable employment demand in the same region.” This does not have to be stated.
100-106. I don’t understand this section.
- Marketization. Please explain in detail, including the unit of measurement.
- What is meant with legitimacy? What is its unit of measurement?
- To promote readability, use GDP rather than REV.
- What is variable SREP?
- If FORI is the same as FDI, use the latter.
207-213. The time span of the study is not clear.
- Table 2. It would be good to show an example how the values in the table were created. Furthermore, what is the meaning of large differences such as the HRSI and Jiangsu vs. Heilongjiang?
- Table 4. For the sake of clarity, explain all the values and cases of the first row.
- Table 6. For the sake of clarity, explain all the values and cases of the first row.
- The path is said to cover nine cases, but only seven are reported.
- The problem of survival: does the central government not financially assist struggling provinces?
389-391. The share of industry and services in less developed provinces is higher than in developed provinces? The statement seems counterintuitive.
- If MOHR is a necessary condition for a boom in HRSI, tell more about it. What kind of public policies are consistent with an increasing MOHR?
455-457. What kind of public policies are consistent with this core configuration?
- How do we know that market-oriented reform is needed? I don’t remember having seen any data on the respective provinces’ market credentials.
- “[…] promote economic upgrading, and form the driving force for HRSI.” Too general, delete.
- Are the interacting and counteracting drivers known? Explain them.
- “[…] not very clearly […].” This sentence undermines the study’s findings. Did the study find something of importance, or not?
442-552. Make sure that the conclusion section lays out clearly the desirable and undesirable development paths, and outlines the concrete measures required to get to the former (to avoid the latter).
Author Response
Dear reviewer,
Thank you very much for your message concerning our manuscript sustainability-1069969. We genuinely appreciate your valuable comments and suggestions. According to the suggestions, the manuscript has been revised carefully.
Comment 1: This is an interesting study with a rarely seen methodology. On the other hand, it leaves many terms undefined, its policy implications and conclusion are weak, and above all, the English language in many parts of the manuscript is hard if not impossible to understand. All three facets need some serious extra work.
Response: Thank you for your systematic review and suggestions. The suggestions help us to improve the readability and scientific value of this article greatly. We modified the context carefully following your suggestions.
Comment 2: Row 9. Human Resource Service (HRS) was not defined and an internet search wasn’t much help either. Since HRS is in the heart of the study, it is critical to explain the concept to the reader in detail.
Response: Thank you for your kind suggestion. To understand the construct better, we have added the definition of HRS in the revision, as shown in the beginning of the abstract and the introduction section in row 9-11 and row 43-47.
Comment 3: Row 10. What is “marketization of human resources” (MOHR)?
Response: Thank you for your comment. This is an important issue to be discussed. The marketization of human resources refers to that both workers and employers participate in the labor market, and allocate human resources through the market rather than government bureaucracy. The labor market is common in lots of countries, while the cultivation of the market is still a work in progress for China and other emerging economies. To demonstrate this construct better, we have added the explanation marked in the revision, as shown in row 19-20 and row 136-147.
Comment 4: Row 22-23. “[…] transforming economies should […] give impetus to economy transformation.” I don’t think anyone argues against this. Delete.
Response: Thank you for your kindly reminding. We have deleted the sentence.
Comment 5: Row 28. Define Human Resource Service clearly.
Response: Thank you for your valuable suggestion. We have defined Human Resource Service clearly in row 43-47 in the revision.
Comment 6: Row 35-36. “[…] promoted the regional employment, but also created numerous jobs.” How else?
Response: Thank you for your comment. We have rewritten the section and present a supplementary argument in row 51-53 in the revision.
Comment 7: Row 75-76. “[…] the more regional employable population, the larger sustainable employment demand in the same region.” This does not have to be stated.
Response: Thank you for your kindly suggestion. we have deleted the sentence.
Comment 8: Row 100-106. I don’t understand this section.
Response: Thank you for your comment. To make the expression clearer, we have rewritten the section in the revision, as shown in 130-146.
Comment 9: Row 107.Marketization. Please explain in detail, including the unit of measurement.
Response: Thank you for your comment. We have explained this construct in detail in the revision, as shown in row 138-148.
Comment 10: Row 108.What is meant with legitimacy? What is its unit of measurement?
Response: Thank you for your comment. We consider that legitimacy in this context refers to social legitimacy, which means the degree of social acceptance for a particular thing. In China, the government serves all sectors of society. Before formulating a public policy, the government generally obtains the views of various sectors of the community. The public policy can better represent the attitudes of the community towards a particular issue. Finally, we have demonstrated the construct clearly in the revision, as shown in row 152-167.
Comment 11: Row 109.To promote readability, use GDP rather than REV.
Response: Thank you for your kindly reminder. We have used GDP in the revision.
Comment 12: Row 110.What is variable SREP?
Response: Thank you for your valuable comment. This construct refers to the quantity of the employable in the human resource market of a special region. The number of workers is relatively stable generally. However, the labor force in China is often mobile from the countryside of less developed regions to the city of developed provinces. To get employed and maintain a balance of work-family, lots of workers go back and forth between their working place and the registered place usually. Therefore, only the population of long-term residents can be a good indicator of the quantity. Finally, we have explained the variable in row 103-107 in the revision, and used the population of long-term residents (PLR) directly rather than SREP.
Comment 13: Row 111.If FORI is the same as FDI, use the latter.
Response: Thank you for your kindly reminder. We have used FDI in the revision.
Comment 14: Row 207-213. The time span of the study is not clear.
Response: Thank you for your comment. The time span of the study is from the end of 2017 to the end of 2019. To demonstrate the time span clearly, we have rewritten the section in the revision, as shown in row 264-273.
Comment 15: Row 245.Table 2. It would be good to show an example how the values in the table were created. Furthermore, what is the meaning of large differences such as the HRSI and Jiangsu vs. Heilongjiang?
Response: Thank you for your valuable suggestion. We have given an example of how we produce the values in the revision, as shown in row 304-312. Moreover, the large differences, such as the HRSI between Jiangsu vs. Heilongjiang, represent the importance of this research. Finally, we have described the cases and explained the meaning of large differences between cases briefly in row 313-318 in the revision.
Comment 16: Row 246.Table 4. For the sake of clarity, explain all the values and cases of the first row.
Response: Thank you for your valuable suggestion. We have explained all the values and cases of the first row at the end of table 2,3, and 4 in the revision.
Comment 17: Row 247.Table 6. For the sake of clarity, explain all the values and cases of the first row.
Response: Thank you for your valuable suggestion. We have explained all the values and cases of the first row at the end of table 5 and 6 in the revision.
Comment 18: Row 248.The path is said to cover nine cases, but only seven are reported.
Response: Thank you for your careful review. We have carefully examined the analysis process and data, and we have confirmed that the path only covers seven cases. We have modified that sentence in the revision.
Comment 19: Row 249.The problem of survival: does the central government not financially assist struggling provinces?
Response: Thank you for your valuable suggestion. China has made outstanding achievements in poverty alleviation and development in poverty-stricken areas, but the measures taken by the central government to solve poverty, such as financial assistance, have modestly improved the living conditions of residents in the short term. There is still a huge gap in regional development. Finally, we have made a demonstration of our opinion in detail in row 492-496 in the revision.
Comment 20: Row 389-391. The share of industry and services in less developed provinces is higher than in developed provinces? The statement seems counterintuitive.
Response: Thank you for your kindly comment. According to the statistics of GDP in China mainland, the output of the agricultural sector is lower than that of other industrial sectors due to the long-term impact of the industrial and agricultural scissors gap policy. Areas dominated by agriculture tend to have lower GDP than those dominated by industry and services. Therefore, the share of industry and services in less developed provinces is much lower than that in developed provinces. To illustrate our point of view clearly, we have rewritten the section in row 507-514 in the revision.
Comment 21: Row 452. If MOHR is a necessary condition for a boom in HRSI, tell more about it. What kind of public policies are consistent with an increasing MOHR?
Response: Thank you for your comment. Our research shows that MOHR is a possible necessary condition for the development of HRS. We examined policies associated with this condition. For example, public policies should be adopted to promote the MOHR. On the one hand, the government needs to pay more attention to labors’ vocational skills training and give assistance to those who have difficulty hunting jobs. On the other hand, the government of China should encourage employers to hire workers through the public human resources market rather than relying on the arrangements of government officials. Furthermore, the policies should focus on credit system construction and strive to eliminate employment discrimination in the human resource market. We have rewritten the section in row 588-596 in the revision.
Comment 22: Row 455-457. What kind of public policies are consistent with this core configuration?
Response: Thank you for your comment. Based on the research, we recommend the government adopting systematic and complementary public policies to promote the development of HRS in the region. For example, it is necessary to consider the resources endowment of the region and focus on attracting foreign investment that matches the endowment. At the same time, it is urgent to enhance the social status of human resource service providers, and encourage the providers to play important roles in attracting talents, improving the professional skills of workers, and promoting the allocation of human resources. Moreover, policies about attracting the FDI should consider the impact of the investment on the exploitation of regional resources and the development of regional economic upgrading rather than merely GDP. We have rewritten the section as you suggested in row 634-644 in the revision.
Comment 23: Row 492.How do we know that market-oriented reform is needed? I don’t remember having seen any data on the respective provinces’ market credentials.
Response: Thank you for your valuable comment. Existing studies have shown that the MOHR is a vital part of market-oriented economic reform in China. Furthermore, the construction of the factor market cannot be separated from the support of the construction of the whole market economic system. The reform of the employment system goes hand in hand with the reform of the market-oriented reform of China’s economy. To explain to the necessity of market-oriented reform, we have added our argument in row 645-659 in the revision.
Comment 24: Row 493.“[…] promote economic upgrading, and form the driving force for HRSI.” Too general, delete.
Response: Thank you for your kindly reminder. We have deleted that sentence.
Comment 25: Row 494. Are the interacting and counteracting drivers known? Explain them.
Response: Thank you for your valuable suggestion. We have explained the interacting and counteracting drivers more clearly in row 621-633 in the revision. For example, a few regions are employing a relatively inferior population and GDP. While, the regions could improve the quality of economic development, and promote the development of HRS on high-quality demand, to realize the quality advantage to make up for the quantity disadvantage of the employable and GDP. Also, the development of HRSI is not necessarily prosperous for those regions with advantages of employable population and regional economy mainly because of that the resource advantage has not been transformed into an efficiency advantage. In other words, the market-oriented resource allocation efficiency is higher than non-market-based resource allocation, and the efficiency advantage is as important as quantity for the growth of HRS.
Comment 26: Row 495.“[…] not very clearly […].” This sentence undermines the study’s findings. Did the study find something of importance, or not?
Response: Thank you for reminding us that our expression is too modest. We have deleted the sentence and demonstrated our contribution more objectively in the context.
Comment 27: Row 442-552. Make sure that the conclusion section lays out clearly the desirable and undesirable development paths, and outlines the concrete measures required to get to the former (to avoid the latter).
Response: Thank you for your kindly suggestion. We have reorganized the conclusion part of the paper and refined the theoretical contribution and policy suggestions of our research in row 588-753 in the revision.
Moreover, we have followed other reviewers’ suggestions and modified the context marked in red in the revision.
Reviewer 3 Report
Tables in the appendix should have sources.
The following references are not referred to in the text:
Please make sure that all references listed in the list are referred to in the text of the article.
Author Response
Dear Reviewer,
Thank you very much for your message concerning our manuscript sustainability-1069969. We genuinely appreciate your valuable comments and suggestions. According to the suggestions, the manuscript has been revised carefully.
Comment 1: Tables in the appendix should have sources.
Response: Thank you for valuable suggestion. we have added the sources of the tables in the appendix in the revision.
Comment 2: The following references are not referred to in the text: Please make sure that all references listed in the list are referred to in the text of the article.
Response: Thank you for your careful and patient review. We have examined the text carefully. There is no omission of references in context. And, we are sure that all references listed are referred to in the text of the article.
Moreover, we have followed other reviewers’ suggestions and modified the context marked in red in the revision.
Round 2
Reviewer 2 Report
Thanks for the revised manuscript. You did an excellent job in improving the readability of the text. One thing I missed was a comparison with previous studies that had used the same methodology. In case there are no such studies, this point is moot. There are still occasional minor issues with English - it would be good to take care of them.
Author Response
Dear reviewer,
Thank you very much for your message concerning our manuscript sustainability-1069969. We appreciate your valuable comments and suggestions. According to the suggestions, the manuscript has been revised carefully again.
Comment 1: Thanks for the revised manuscript. You did an excellent job in improving the readability of the text.
Response: Thank you for your recognition. We hope the following responses and modifications in the revision would clarify the issues reasonably.
Comment 2: One thing I missed was a comparison with previous studies that had used the same methodology. In case there are no such studies, this point is moot. There are still occasional minor issues with English - it would be good to take care of them.
Response: Thank you for your advice and suggestions for further modifications. As far as we know, there has been no comparable study in the research field of HRS or HRSI.It is a new exploration to study HSR using the method of QCA. Despite this, we try to have a dialogue with those studies that are related to our study. We have supplemented those conversations in row 530-533,565-568, and 610-614 respectively in revision 2.0. These dialogues further illustrate the theoretical innovation of this context.
Moreover, we invited two experts who are familiar with English writing to revise the language of this context. The English language has been checked carefully and necessary changes were made in the revision 2.0.
This manuscript is a resubmission of an earlier submission. The following is a list of the peer review reports and author responses from that submission.